# Chitosan/Polyvinyl Alcohol-Based Biofilms Using Ternary Deep Eutectic Solvents towards Innovative Color-Stabilizing Systems for Anthocyanins

**DOI:** 10.3390/ijms25116154

**Published:** 2024-06-03

**Authors:** Hiléia K. S. de Souza, Marta Guimarães, Nuno Mateus, Victor de Freitas, Luís Cruz

**Affiliations:** 1REQUIMTE/LAQV, Chemistry and Biochemistry Department, Faculdade de Ciências, Universidade do Porto, Rua do Campo Alegre, 687, 4169-007 Porto, Portugal; hileiak@yahoo.com (H.K.S.d.S.); marta-guimaraes1@live.com.pt (M.G.); nbmateus@fc.up.pt (N.M.); vfreitas@fc.up.pt (V.d.F.); 2PIEP—Pólo de Inovação em Engenharia de Polímeros, Universidade do Minho, Campus de Azurém, Edifício 15, 4800-058 Guimarães, Portugal

**Keywords:** ternary deep eutectic solvents, anthocyanins, biopolymers, color stability, water vapor permeability, scanning electron microscopy

## Abstract

Anthocyanins are amazing plant-derived colorants with highly valuable properties; however, their chemical and color instability issues limit their wide application in different food industry-related products such as active and intelligent packaging. In a previous study, it was demonstrated that anthocyanins could be stabilized into green plasticizers namely deep eutectic solvents (DESs). In this work, the fabrication of edible films by integrating anthocyanins along with DESs into biocompatible chitosan (CHT)-based formulations enriched with polyvinyl alcohol (PVA) and PVA nanoparticles was investigated. CHT/PVA-DES films’ physical properties were characterized by scanning electron microscopy, water vapor permeability, swelling index, moisture sorption isotherm, and thermogravimetry analysis. Innovative red-to-blue formulation films were achieved for CHT/PVA nanoparticles (for 5 min of sonication) at a molar ratio 1:1, and with 10% of ternary DES (TDES)-containing malvidin-3-glucoside (0.1%) where the physical properties of films were enhanced. After immersion in solutions at different pH values, films submitted to pHs 5–8 were revealed to be more color stable and resistant with time than at acidic pH values.

## 1. Introduction

Anthocyanins are water-soluble pigments responsible for a wide pallet of appellative colors found in flowers, vegetables, and fruits. The outstanding chemistry behind their physicochemical and biological properties discovered throughout the years has increased the interest in the application of these molecules in food, cosmetic, energy, and biomedical industries [1,2]. On the other hand, several issues mainly related to the chemical and color stability of pH, temperature, and light of these dyes are still big scientific challenges to overcome. That is why several methodologies have been raised over the last years to modulate, promote, and explore novel technological applications. Several examples are reported in the literature dealing with the production of novel stable anthocyanin derivatives either by chemical or enzymatic processes [3], and the bio-inspired synthesis of new anthocyanin-based photosensitizers for energy [4,5] and biomedical applications [6]. More recently, the scientific community has been putting efforts into studying more deeply these compounds at the molecular level to find novel biocompatible approaches and green strategies to address novel solutions to increase their stability [7,8,9], including the development of host-guest and self-assembled systems [10,11], polyelectrolyte complexes (PECs) [12], and anthocyanin-based films [13].

Moreover, micro- and nano-systems for the chemical/color stabilization of anthocyanins have been developed in recent years mainly including nanoemulsions, liposomes, biopolymer-based nanoparticles, nanogel, complex coacervates, among others [14,15,16,17]. Chitosan (CHT) is a marine-origin polysaccharide extracted from shrimps with biomedical interest in the development of bioactive films due to its excellent biocompatibility and antibacterial properties. Furthermore, polyvinyl alcohol (PVA) is a popular water-soluble polymer widely used in the food packaging industry because of its high chemical and thermal stability, high strength and high optical transparency in water, and low manufacturing cost. CHT/PVA films containing anthocyanins have been already reported in the literature mainly as label freshness indicators for food smart packaging [18,19,20,21]; however, the fabrication of PVA nanoparticles to enhance the stability of anthocyanins in CHT films was never explored. On the other hand, deep eutectic solvents (DESs) have been rising since the early 2000s as a sustainable and eco-friendly alternative to classical organic solvents for example for extraction of anthocyanins [22,23]. DESs are defined as homogeneous eutectic mixtures obtained by mixing two or more pure components (liquids or solids, or ions or neutral molecules) acting as hydrogen bond acceptors (HBA) and hydrogen bond donors (HBD). The main feature of DESs is their tunability to pair a large number of diverse HBA and HBD in different molar ratios that allows the preparation of DESs with specific properties, as well as their low toxicity, low volatility, and flammability [24].

In our previous work, the development of anthocyanin formulations in binary and ternary choline chloride-based DESs was achieved and their structural, chemical, morphological, and rheological properties were deeply studied by several techniques [8]. Overall, the red color stabilization of anthocyanins was favored in choline chloride/glycerol/ethylene glycol (ChCl/Gly/EG) ternary mixtures. Most of the works, including anthocyanin-based films for food smart packaging, did not correlate the pH-dependent color variation with their chemical species existing in the equilibrium network.

Bearing this, and based on our previous knowledge, in this work the main goal was to develop CHT/PVA nanoparticles-based films containing anthocyanins directly dissolved in ternary DES-based mixtures in order to act directly as plasticizers and to promote the stability of the red- and purple-blue-colored species of anthocyanins (flavylium cation and quinoidal bases, respectively) on solid materials. The CHT/PVA-DES films were characterized by scanning electron microscopy (SEM), water vapor permeability (WVP), swelling index (SI), moisture sorption isotherm, thermogravimetry analysis (TGA), and the CIELAB color properties.

## 2. Results and Discussion

### 2.1. DLS

The sonication assays for the formation of the PVA nanoparticles were studied over time and the data (Z-average and PDI) were analyzed for each point. Figure 1 shows the Z-average diameters (according to an ISO method ISO 22412:2008 [25]) of both the non-sonicated and sonicated PVA solutions with sonication time varying between 0 and 15 min. The results obtained from the DLS analysis indicate that an increase in sonication time tremendously reduces the size and width of the PVA particles. 

The DLS analysis shows that the Z-average diameter values of the sonicated PVA solution, depending on the time process, ranged from 244 (non-sonicated PVA solution) to 70 nm (PVA solution submitted at 15 min of sonication).

Moreover, the polydispersity index (PDI) values for both the non-sonicated and sonicated solutions increase by increasing sonication time (from 0.3 PDI without sonication to 0.6 PDI after 2 min of sonication) indicating the existence of a polydisperse and non-stable particle distribution as the result of particle aggregation when the PVA solution is sonicated.

The 3D networks of the non-sonicated (0 min) and sonicated PVA solutions (5 min) were observed by cryoSEM imaging (Figure 1) where the non-sonicated PVA solution showed a better-defined polymeric network. The opened-polymeric network obtained for PVA sonicated for 5 min (SP_5_) seemed to confirm the positive contribution of sonication time to the assembly formation (aggregation) producing a smaller network.

### 2.2. Density and Viscosity

Figure 2 presents the density values of the non-sonicated and sonicated PVA solutions prepared using different ultrasound treatment times. The density of the PVA solution without the ultrasound treatment was the lowest, while those of the sonicated solutions increased when the solution was exposed at different ultrasonication times. The results show that when using an ultrasonication time of 2 min, the density of the solution increased significantly. More ultrasonication time results in solutions with similar density values. The trend agrees with the PdI tendencies shown in Figure 1. This is probably due to the combined effects of the amount of free hydroxyl groups and the different particle sizes inserted within the sonicated PVA network.

The viscosity of the PVA solution decreased after 2 min of ultrasonication (Figure 2). Increasing sonication to 5 min leads to a significant drop in the viscosity. These results agree with the results published by Abral et al. (2020) [26].

According to the authors, the depolymerization followed by the reduction in the PVA molecular weight when the solution is submitted to the ultrasonication process can explain the viscosity behavior.

### 2.3. Refractive Index

The refractive index values of materials like PVA and CHT are important for various applications, including optics, material characterization, and quality control [27,28]. The refractive index of PVA can vary depending on factors like its degree of polymerization, molecular weight, and degree of hydrolysis. On the other hand, CHT has a refractive index that can also vary depending on its source, degree of deacetylation, and molecular weight. The refractive index (n) values of the PVA samples (submitted at different times of sonication), CHT solution, and DES were obtained. No significant change in refractive index has occurred for PVA solutions at any time of sonication. For CHT, a slight increase in refractive index was registered whereas higher refractive index values for the DES solutions were determined (Table 1).

### 2.4. Chitosan Intrinsic Viscosity

The intrinsic viscosity [*η*] of chitosan was evaluated at 25 °C and pH 4.7 using the Huggins and the Kraemer equations described elsewhere [29]. As can be seen in Table 2, the two extrapolations gave similar results of [*η*].

### 2.5. Cryo-SEM

Before the process of drying down to films, where significant structural changes take place, the cryo-SEM approach enables the viewing and morphological characterization of coacervate structures in their original environment. Figure 3 displays the cryo-SEM findings: distinct network structure, mesh, layout, and bonding in the solution, which correlate to distinct morphologies, were undoubtedly caused by the variable solution composition. Overall, it seems that with 10% of TDES, the polymeric network is more compact and organized whereas with 20% some parts of the network start to be broken with the appearance of more empty spaces. The presence of anthocyanin in the solution did not alter the polymers’ morphological structure.

### 2.6. SI

We examine the swelling behaviors of the CHT films mixed with PVA and PVA-NPs (Figure 4A–D). For all the systems (with or without PVA nanoparticles), the SI increases with the increasing of immersing time until it reaches a plateau. This behavior can be clarified by recognizing that the polymer blend’s interconnected chains, formed through crosslinking, undergo decreased mobility during extended hydration. This limited mobility hinders the accessibility of the solvent, consequently impacting the hydration dynamics of the film [21]. In general, PVA and PVA-NP systems present higher swelling indexes.

Overall, the CHT/PVA films showed an intermediary SI % between the neat CHT and neat PVA films, except for the films obtained with CHT/PVA_5_-NPs (1:1) for 5 min which have the highest SI compared to the correspondent NADES that present the lowest swelling index.

Due to the abundant hydroxyl groups, PVA demonstrates heightened hydrophilicity in contrast to CHT. Conversely, CHT contributes to a reduction in the film’s hydrophilicity through the integration of hydroxyl and acetate groups, enabling the formation of intra- and inter-molecular hydrogen bonds [21]. Consequently, the ratio (1:1) between CHT/PVA_5_-NPs could be described as an ideal ratio to employ owing to its superior swelling capacity when compared to the precursors.

Overall, the presence of mv3glc in the film increased the SI, which could be related to the hydrophilic character of this kind of water-soluble pigment that might promote water entrapment within the polymeric matrix (Figure 5).

Regarding the stability of the films when immersed for 3 h in water at different pH values, it could be observed that for higher pH values, the films were much more resistant rather than at acidic pH (Figure 6). Actually, at more acidic pH values, the films rapidly start to dissolve mainly because of the higher solubility of CHT and the presence of the anthocyanin dye at hydrophilic flavylium cation species. Interestingly, the high color stability and the low SI were obtained for the films at a more basic pH which is an innovative and relevant result since the stabilization of the quinoidal base species of anthocyanins in solution is difficult to achieve.

### 2.7. WVP

Since it directly impacts other factors, thickness is a significant element for research on the physical characteristics of films as well as mechanical properties or physicochemical characteristics like WVP.

Regarding the WVP results (Figure 7a), the compromise of obtaining the lowest water vapor permeable film with an equilibrated PVA nanoparticles size was obtained using the conditions of 5 min of sonification for CHT/PVA 1:1. This film was chosen considering the next studies of color stability with the pH of the medium.

Regarding the effect of adding different percentages of TDES aiming to act both as a plasticizer and a solvent for anthocyanins (Figure 7b), it was observed that the WVP values of CHT/PVA 1:1 films increased with the TDES percentage which could take to a more widely and opened polymeric network.

Figure 7c displays the WVP values obtained for the films fabricated with different anthocyanin structures as well as when submitted to different pH environments. The films fabricated with extracted malvidin-3-glucoside revealed an inferior WVP value than the one with the respective aglycone obtained by synthesis (3-deoxymv) as well as that with extracted cyanidin-3-glucoside.

Moreover, the pH variation could modulate the WVP of the film depending on the goal: reducing it in order to improve the barrier properties taking into account an active packaging perspective, or increasing it in order to produce pH/freshness smart labels for the real-time control of food spoilage. In this case, the trend observed was that the lowest WVP value was obtained for the film with the highest pH which could be related to the high hydrophobicity of the quinoidal base species of anthocyanins presented at high pH and the precipitation of the CHT polymer as well.

### 2.8. Moisture Sorption Isotherms

The water sorption characteristics of the films are intricately linked to the microstructure of the matrix and are influenced by the relative humidity in the environment. The higher solubility, water affinity, and water retention of the films result in higher moisture content (*X_e_*) at a given water activity (*a_w_).*

The moisture sorption isotherm curves, *X_e_* in the function of *a_w_*, *f* or the studied PVA films (NS, PVA_3_-NPs, PVA_5_-NPs, and PVA_7_-NPs), CHT films, and CHT/PVA blend films have been illustrated in Figure 8 and Figure 9. As can be observed in the figures, the adsorption isotherm profiles of the examined films depict the changes in moisture content concerning water activity (*a_w_*). The outcomes indicate that these responses exhibit a type III sigmoidal shape, which aligns with the classification described in the literature [30].

Some authors have noted that moisture sorption isotherms reflect the collective hygroscopic properties of the individual components within the films [31]. In the present study, the films containing PVA nanoparticles exhibited similar behavior when exposed to varying relative humidity, except for the CHT/PVA_7_-NPs (7:3) film (Figure 8D) and *CHT/PVA_5_-NPs* (1:1) with different TDES plasticizer contents (*w*/*w*) (Figure 9) which showed a decrease in relation to the others, suggesting less hygroscopicity. As evident from the data, the equilibrium moisture content exhibited a nearly linear increase up to water activity (*a_w_*) until 0.6. Beyond this point, the rate of increase became exponential. This non-linear sorption profile is a characteristic feature of hydrophilic films and has also been observed in other CHT-based film studies [32]. In general, the films with TDES show a lower capacity to adsorb water at high water activities, while the films without plasticizers adsorb more at lower water activities.

The GAB model’s wide acceptance in the field of food science and materials is attributed to its unique ability to calculate the monolayer moisture content, providing insight into the amount of water absorbed by specific sites on the surface of food. This crucial parameter plays a pivotal role in maintaining food stability by guiding researchers in determining the optimal moisture content for various food products. Furthermore, the GAB model stands out for its consideration of multiple layers and condensed film water, and each constant within the model holds a well-defined physical meaning, making it a valuable tool in the study of food properties and stability.

Table 3 provides the GAB model parameters (*X_0_*, *C*, *k*). The values of the determination coefficient (*R^2^ ≥* 0.988) and the *k* parameter (0 < *k* < 1) demonstrate that the GAB model is an effective tool for accurately fitting the experimental data.

The estimated GAB parameters vary significantly and depend on factors such as the concentration of CHT, the type of PVA (NS or PVA_n_-NPs), the presence of a plasticizer and at different pH conditions (Table 3, Table 4 and Table 5).

The Guggenheim constant, C, is found to be lower for CHT/PVA_5_-NP 1:1 films without a plasticizer. These lower values of the C parameter indicate that the water sorption is primarily characterized by a monolayer of water molecules that are not strongly bound to the polymer at the primary sorption sites. This suggests that subsequent water molecules tend to form more structured layers in the upper layers of the material, and a more organized multilayer structure can be anticipated.

*X*_0_, which represents the amount of water retained at the primary sorption sites of the biobased films on a dry basis, is an important parameter and was quantified. In general, pure PVA and pure CHT films have significantly lower *X*_0_ values compared to the CHT/PVA biobased film. No linear correlation between *X*_0_ values and the content of PVA nanoparticles in the material was observed (Table 3).

Interestingly, the plasticized CHT/PVA_5_-NP 1:1 20% TDES films exhibit higher X_0_ values (3.641 in Table 4) compared to their non-plasticized counterparts (0.164 in Table 3). This phenomenon can be attributed to a decrease in the availability of active water sorption sites, which results from the chemical and physical alterations induced by changes in polymer concentration [33]. These observations are consistent with the reduced water vapor permeability (WVP) values noted for these films, as evidenced in the WVP data. Furthermore, this may suggest an improved capacity for water uptake in the monolayer of the plasticized films, particularly under equilibrium conditions.

**Table 5 ijms-25-06154-t005:** GAB parameters obtained from the fitting of the data displayed in Figure 10 to the GAB equation for the biobased films prepared from CHT/PVA_5_-NPs (1:1) with 10% TDES in the presence of mv3glc and at different pH conditions.

	pH	*d* (mm) × 10^−2^	C	*k*	*X* _0_
CHT/PVA_5_-NPs	neat	2.59 ± 0.20	0.007	1.002	0.823
	8	3.28 ± 0.21	0.037	0.889	0.368
	6	3.37 ± 0.18	0.009	0.950	1.881

### 2.9. TGA

The thermal decomposition behavior of the PVA_5_-NP system is illustrated in Figure 11, revealing initiation around 190 °C with two distinct phases of mass loss: the first stage of mass loss has a melting temperature (T melting) of 251 °C, while the second is 452 °C. These findings are quite different from the results published by Gomaa et al. (2018) [34] and Mohamed et al. (2018) [35]; however, there are similarities with the results reported by Peng et al. (2007) [36] where the initial degradation step corresponds to the thermal decomposition of hydroxyl groups, leading to the generation of reactive radicals through elimination reactions. Conversely, as the temperature rises, chain scission and cyclization reactions become prominent in the second melting phase. This process results in the gradual and complete decomposition of hydroxyl groups.

When CHT and/or mv3glc are present, the TGA curves exhibit three distinct weight loss regions. Intriguingly, when these components are combined with PVA_5_-NPs, the stability is enhanced, evident from the fact that the initial decomposition for both systems begins at 212 °C. The first region, spanning temperatures between 50 and 212 °C, is attributed to the loss of absorbed water molecules. The second and third regions mirror those observed in PVA_5_-NPs, signifying the loss of water associated with the polymer matrix and the subsequent decomposition and carbonization of the polymer. It is noteworthy that the residue values for both blends are notably higher (40%) compared to the precursor (20%). Also looking at Figure 11, it is demonstrated that the CHT/PVA mixtures in the presence of mv3glc presented higher thermostability than with cy3glc.

### 2.10. SEM and pH-Dependent Chromatic Properties

The films of CHT/PVA_5_-NPs 1:1 with 10% TDES and with mv3glc were immersed in water for 3 h at different pH values from acid to basic (Table 6). It was verified that after being immersed in acidic solutions (pH 3, 4), the films rapidly solubilized and degraded, and therefore, were not robust enough. At high pH values, the films start to be more water resistant and with intense red to blue colors. It is noticeable that the a* axis relative to the green-magenta opponent colors becomes more negative toward green with the pH increase, and the b* axis represents the blue-yellow opponents where all the films revealed negative values which are consistent with their blue color appearance.

The most promising CHT/PVA_5_-NP 1:1 + 10% TDES + mv3glc films exposed at pH values of 5.7, 6, and 8.2 were kept in desiccators with different water activities (a_w_) and their CIELAB color coordinates measured over time (Table 7). After 19 days, the b* parameter of the films at pH 5.7 significantly increased towards the yellow axis as long as the a_w_ values also increased. On the other hand, the films at pH 6 were measured after 7 and 70 days and, in general, they were revealed to be more purple-blue stable over time for the different a_w_ environments studied. The most interesting result was obtained for the film at pH 8.2 which demonstrated the highest stability after 70 days of storage in all a_w_ environments. The presence of a superior mole fraction of the quinoidal base species (neutral and anionic) of mv3glc at this pH should be a key factor for obtaining this color-stabilizing system.

## 3. Materials and Methods

### 3.1. Materials

Malvidin-3-glucoside (mv3glc) and cyanidin-3-glucoside (cy3glc) were obtained by extraction from a young red wine (*Vitis vinifera* L. cv., Touriga Nacional) and blackberries (*Rubus fruticosus* L.), respectively, as described elsewhere [37,38].

Briefly, cy3glc was extracted from frozen blackberries with MeOH:H_2_O 1:1 *v/v* (0.1 M HCl) for 1 h at RT. After that period, the extracted juice was filtrated using a nylon membrane and MeOH was eliminated by evaporation. The aqueous fraction was purified with a Buchner funnel loaded with reverse-phase C18 silica gel where cy3glc was isolated with 10% aqueous MeOH (0.1 M HCl). After MeOH evaporation, the cy3glc fraction was lyophilized and stored at −18 °C until use.

Briefly, mv3glc was isolated from a young red wine previously concentrated in a nanofiltration system to eliminate ethanol whereas sugars were removed by reverse-phase C18 silica gel. The aqueous fraction was extracted with ethyl acetate to remove non-anthocyanin organic compounds. The aqueous layer was then added in a Buchner funnel loaded with TSK Toyopearl gel HW-40(S) and mv3glc was eluted with 10% aqueous MeOH acidified with 0.1 M HCl. After MeOH evaporation, the mv3glc fraction was lyophilized and stored at −18 °C until use.

3-Deoxymalvidin (3-deoxymv) was synthesized through an acid-catalyzed aldol condensation between 2,4,6-trihydroxybenzaldehyde and 4-hydroxy-3,5-dimethoxyacetophenone according to the procedures described in the literature [39]. Chitosan (CHT) powder from ChitoClear (Mw = 150–250 kDa, 78% degree of deacetylation) was purchased from Primex ehf (Siglufjordur, Iceland). Polyvinyl alcohol (PVA), choline chloride (ChCl), glycerol (Gly), and ethylene glycol (EG) were obtained from Sigma-Aldrich and used as received.

### 3.2. Preparation of PVA and CHT Solutions

PVA starting mixture (2.0% *w*/*w*) was obtained by the dispersion of PVA in acetate buffer solutions (0.1 M, pH 6) containing sodium azide (NaN_3_) solution in buffer (0.02% *w*/*w*). The dispersion was heated at 80 °C and moderately stirred for 2 h until total solubilization.

A CHT stock solution (2.0% *w*/*w*) was prepared by dispersing CHT in acetate buffer solutions (0.1 M, pH 6) with NaN_3_ (0.02% *w*/*w*) and centrifuged (21,000× *g*, 25 min, 25 °C).

### 3.3. Preparation of PVA Nanoparticles

The PVA nanoparticles (PVA-NPs) were fabricated by ultrasonication based on the method described in the literature [26]. Briefly, 45 g of a PVA aqueous solution (2% (*w*/*w*)) prepared as described above was placed in ice and exposed to ultrasonic irradiation for defined periods of time (degradation time, t_S_ 1, 2, 3, 5, 7, 10, or 13 min) using an ultrasonic probe (model Bandelin Sonoplus D-12207 Berlin, Germany), equipped with a ½ microtip (70% potency, 3 cycles). The samplings were performed in triplicate.

### 3.4. Preparation of Solutions

The film-forming solutions were obtained by mixing non-sonicated PVA (PVA-NS) or PVA-NPs with CHT solutions. The mixtures were obtained by adding the PVA content (0, 30, 50, and 70%) to CHT solutions. The appropriate quantities of plasticizer (TDES of ChCl/Gly/EG 1:1:1) were weighted (0 and 30 wt.% plasticizer per dried PVA amount).

Regarding the anthocyanin-containing solutions, the anthocyanin powder was first dissolved in a CHT solution while constantly stirring. After the complete solubilization of anthocyanin, an appropriate amount of PVA solution was added to the solution to achieve a CHT/PVA ratio of 1:1 where the final mixture was mixed for 10 min.

### 3.5. Fabrication of Bioplastic Films

All the films (PVA, CHT, and CHT/PVA with or without anthocyanins) were obtained by solvent casting method. For this purpose, 20 g of the solution was spread into Petri dishes (8 cm diameter). The films were dried at room temperature for approximately 24 h. After that period, they were detached from the supports, and homogenous films were obtained in all the cases studied.

### 3.6. Characterization of PVA Nanoparticles by DLS

The particle size (hydrodynamic diameter, as Z-Average) and the polydispersity index (PDI) were measured over time by DLS in a Malvern Zetasizer Nano ZS instrument (Malvern instrument Ltd., Malvern, UK). The intensity of light scattered at an angle of 173° was measured by an avalanche photodiode after the solution was illuminated by a 633 nm laser. The aliquots of the solutions (sonicated and non-sonicated PVA solutions) were diluted 150 times in polystyrene zeta cuvettes (10 mm optical path). After equilibration for 300 s, the samples were measured at least 16 sub-run times per analysis run, and each sample was read in triplicate.

### 3.7. Viscosity, Refraction Index, and Density

The viscosity (*η)* of CHT solutions was determined in an Anton Paar viscosimeter model Lovis 2000 ME (Anton Paar, Tokyo, Japan) through the multi-concentration method and the experimental conditions were followed as described elsewhere [40].

The intrinsic viscosity [*η*] (obtained by both Huggins and Kraemer plots), was used to estimate the viscosity-average molecular mass (*M_V_*: Da) through the Mark/Houwink/Sakurada expression (Equation (1)):(1)$$\left[\eta \right]=K{\left[M_{V}\right]}^{a}$$
where *K* and *a* are constants for a given solute/solvent system obtained from Kasaai (2007) [41].

### 3.8. Cryo-SEM

The cryo-SEM/EDS exam was performed using a High-Resolution scanning electron microscope with X-Ray Microanalysis and CryoSEM experimental facilities: JEOL JSM 6301F/Oxford INCA Energy 350/Gatan Alto 2500, Tokyo, Japan. The specimen was rapidly cooled (plunging it into sub-cooled nitrogen—slush nitrogen) and transferred under vacuum to the cold stage of the preparation chamber. The specimen was fractured and sublimated (‘etched’) for 120 s. at −90 °C, and coated with Au/Pd by sputtering for 45 s. The sample was then transferred into the SEM chamber. The sample was studied at a temperature of −150 °C.

### 3.9. Films Characterization

#### 3.9.1. SEM

The morphological characterization of the films was performed in a scanning electron microscope FEI Quanta 400 FEG (FEI, Hillsborough, OR, USA), under a high vacuum using the conditions described elsewhere [42].

#### 3.9.2. WVP

WVP was calculated as described elsewhere [43], in accordance with the Standard Test Method for Water Vapor Transmission of Materials [44]. The film samples with 8 cm diameter were tightly sealed to a permeation cell containing an anhydrous salt (calcium chloride, R.H. = 2%) after being left to equilibrate at 25 °C and 53% R.H. for at least 72 h. The cells were placed in a desiccator with air convection to promote water diffusion, at constant temperature (25 °C) and R.H. (100%). The weight of the cells was registered periodically and the water vapor permeability (WVP), in g∙m^−1^∙s^−1^∙Pa^−1^, of each film was calculated from Equation (2):(2)$$\mathrm{WVP}=\frac{\Delta m\cdot x}{A\cdot \Delta t\cdot \Delta p}$$
where Δ*m* is the weight gain (g), *x* is the film thickness (m), and *A* is the area (0.003 m^2^) exposed for a time Δ*t* (s) to a partial water vapor pressure Δ*p* (Pa).

#### 3.9.3. SI

The SI of the films was defined as the water sorption capacity by the dry film after 24 h of immersion in water [45,46]. The films previously dried in an oven (100 °C for 24 h, *W*_1_) were immersed in 20 mL of Milli-Q water under constant agitation at 150 rpm using a mechanical shaker for 24 h at 25 ± 2 °C. The swollen samples were removed, and the weight was recorded as wet weight (*W*_2_). SI (%) was calculated as an average of three replicates according to Equation (3) [47]:(3)$$\mathrm{SI}\left(\%\right)=\frac{W_{2}-W_{1}}{W_{1}}\times 100$$

#### 3.9.4. TGA

The thermal stability studies of the solid samples (~4 mg) were performed in STA7200RV from Hitachi Instruments, Inc. (Tokyo, Japan) according to the procedures described elsewhere [8].

#### 3.9.5. Moisture Sorption Isotherm

Water sorption isotherms were obtained gravimetrically using the method previously reported [42].

The Guggenheim/Anderson/de Boer (GAB) model was used to represent the experimental sorption data.
(4)$$X_{e}=\frac{CkX_{0}a_{w}}{\left[\left(1-ka_{w}\right)\left(1-ka_{w}+Cka_{w}\right)\right]}$$
where $X_{e}$ represents the equilibrium moisture content, $a_{w}$ is the water activity, $X_{0}$ is the monolayer moisture content, C is the Guggenheim constant, and k is the corrective constant.

### 3.10. Statistical Analyses

Statistical analyses were conducted using ANOVA with a Bonferroni post-test via the GraphPad Prism software, version 5.00 for Windows (GraphPad Software, Inc., San Diego, CA, USA). Significance was determined at *p* < 0.05 to assess mean values and differences among them.

## 4. Conclusions

Anthocyanin-based films containing mixtures of CHT/PVA have been widely reported in the literature for application in food packaging. Herein, a deep study of the effect of PVA sonification time to produce nanoparticles to formulate with CHT dispersions was carried out. Furthermore, the use of TDES-containing anthocyanins as plasticizers to directly produce functional films was also considered. Overall, innovative films with both improved physical/chemical properties and chemical/color stabilization of anthocyanins were obtained mainly at neutral to moderate basic pH environments. The films produced with mv3glc revealed better robustness in terms of general physical characteristics than the ones made with cy3glc.

## Figures and Tables

**Figure 1 ijms-25-06154-f001:**
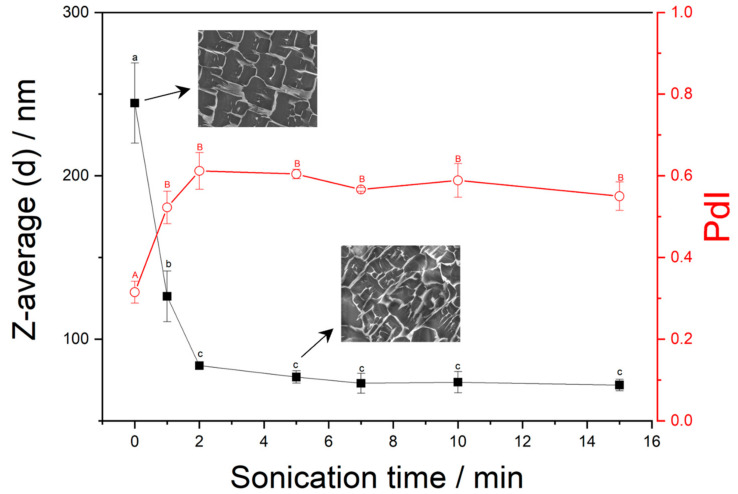
Z-average and PdI values of the PVA nanoparticles as a function of sonication time and cryo-SEM images (2000×) for the non-sonicated PVA (0 min) and sonicated PVA (5 min). ^a,b,c,A,B^ data with the same letters are statistically similar at a 95% confidence level.

**Figure 2 ijms-25-06154-f002:**
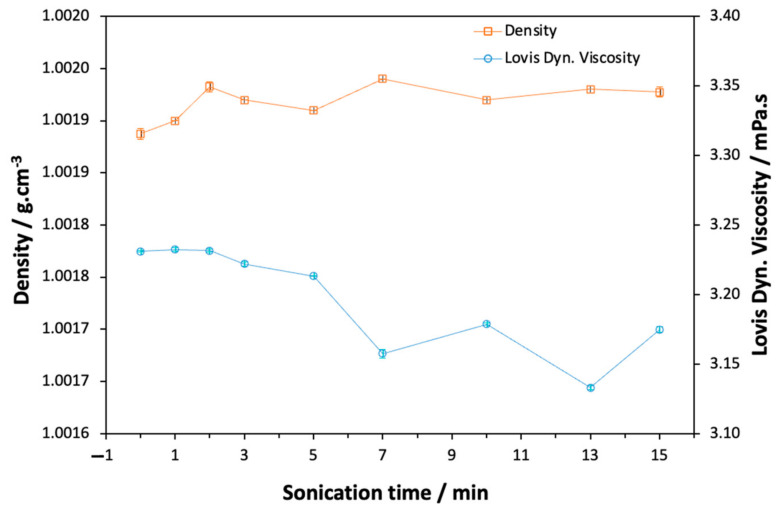
Variation in the density and viscosity of the PVA solutions with the sonication time.

**Figure 3 ijms-25-06154-f003:**
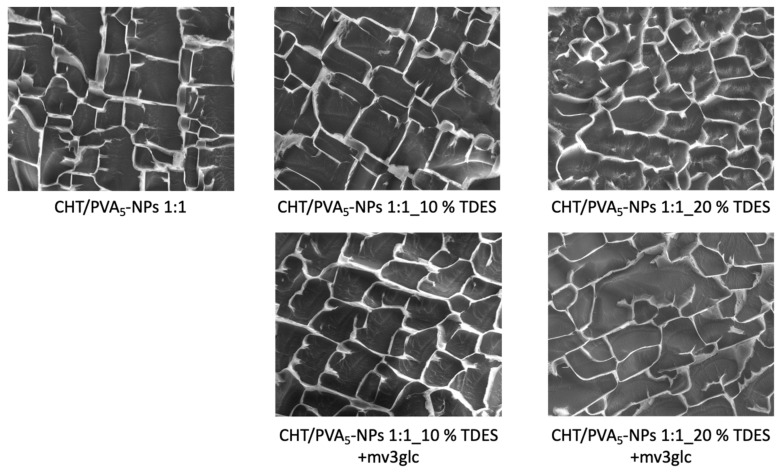
Cryo-SEM images of CHT/PVA_5_-NP 1:1 mixtures (2000×) in the presence and absence of TDES (10, 20%) and anthocyanin content.

**Figure 4 ijms-25-06154-f004:**
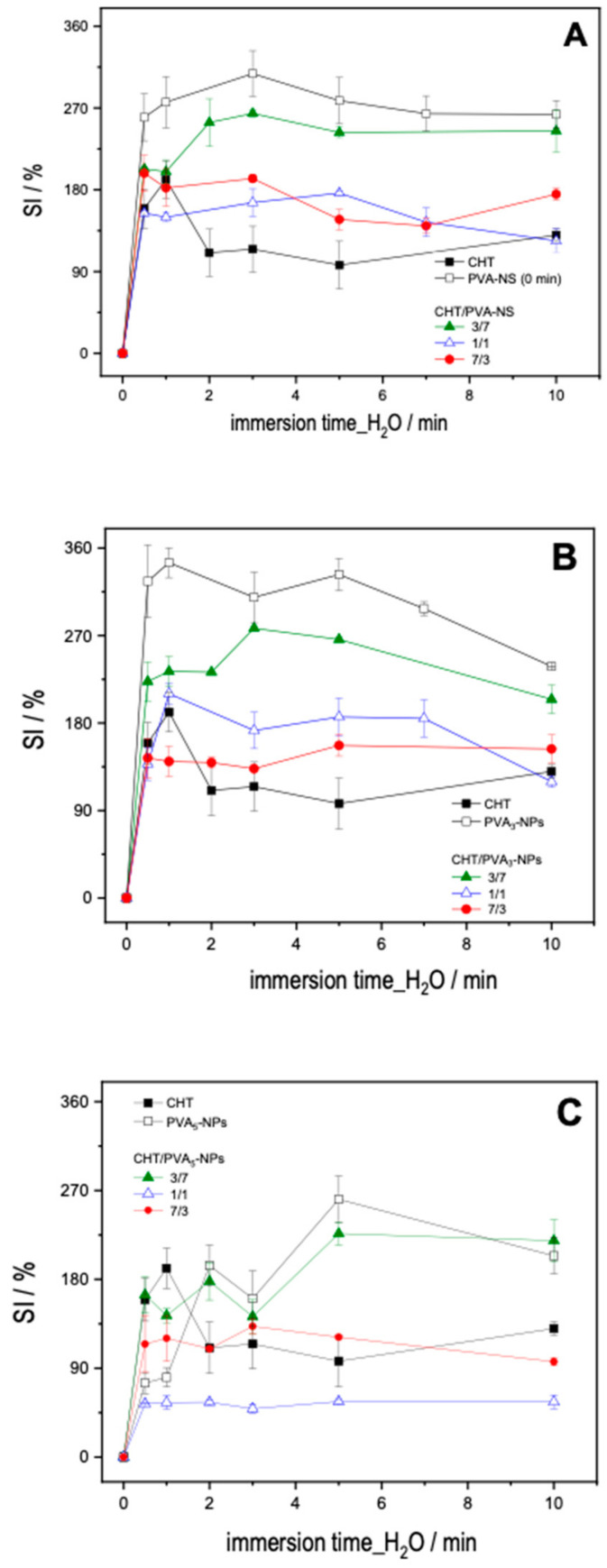
SI of the films obtained after H_2_O immersion as a function of time: (**A**) 0 min of sonication, PVA-NS; (**B**) 3 min of sonication, PVA_3_-NPs; (**C**) 5 min of sonication, PVA_5_-NPs; (**D**) 7 min of sonication PVA_7_-NPs. Molar ratios for the CHT/PVA-NS or CHT/PVA_n_-NPs mixtures were 3/7, 1/1, and 7/3.

**Figure 5 ijms-25-06154-f005:**
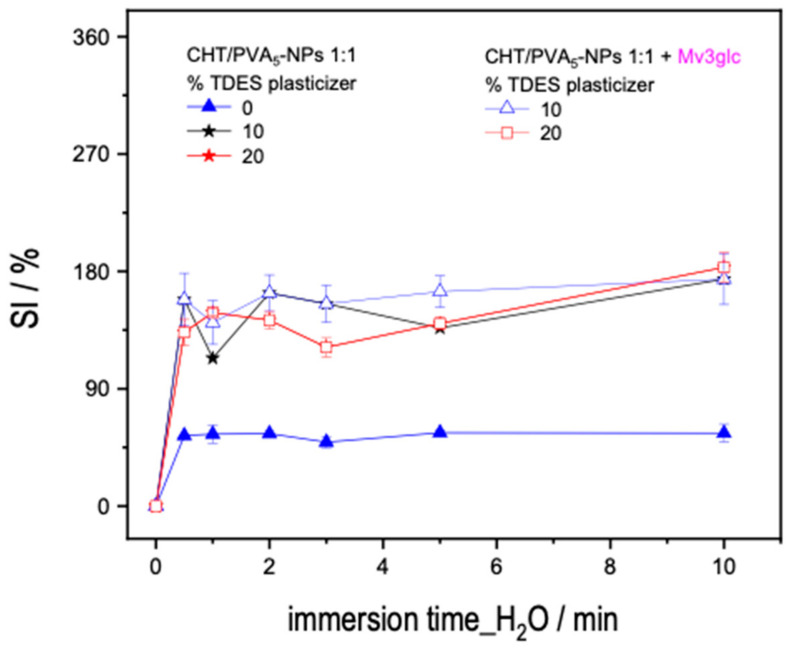
SI of the CHT/PVA_5_-NP 1:1 films containing different percentages of TDES plasticizer with and without mv3glc obtained after H_2_O immersion as a function of time.

**Figure 6 ijms-25-06154-f006:**
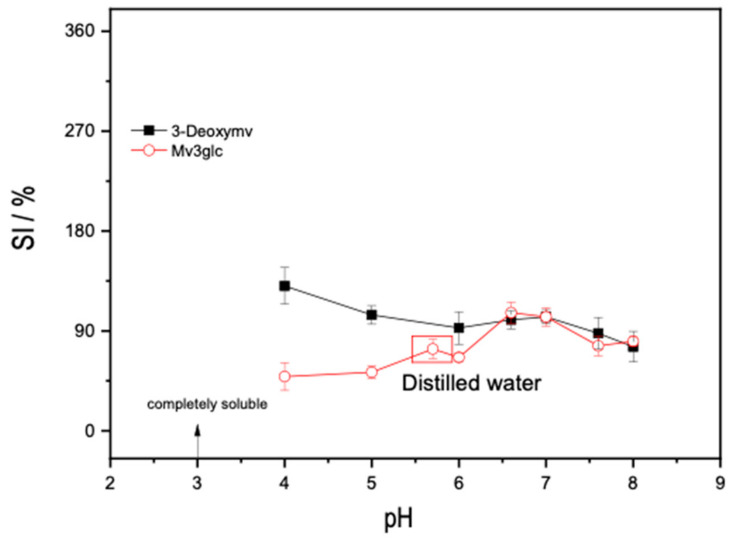
SI of anthocyanin-containing films obtained after 3 h of immersion in water solutions at different pH values.

**Figure 7 ijms-25-06154-f007:**
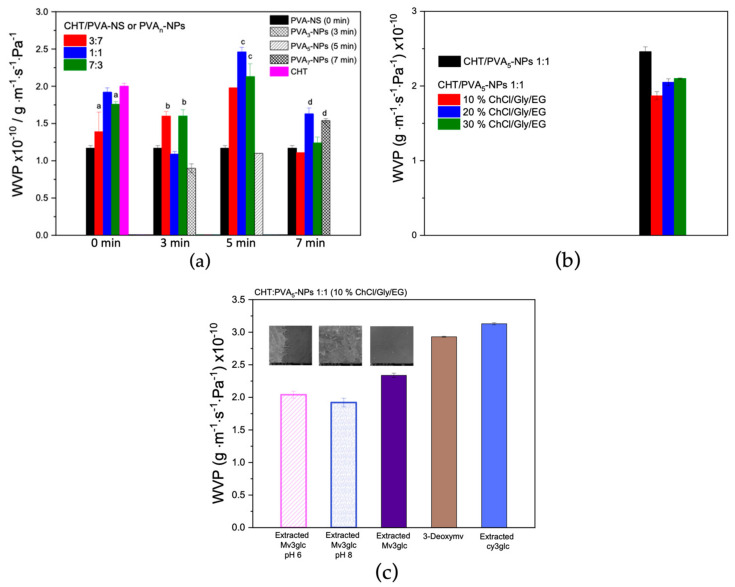
(**a**) WVP results for the films obtained with CHT/PVA-NS at different ratios and at different sonication times (CHT/PVA_n_-NPs) compared with the controls, non-sonicated PVA (PVA-NS 0 min), and CHT; (**b**) the effect of the different TDES percentages on the WVP results of the films; (**c**) the effect of the different anthocyanin structures in the WVP results and cryo-SEM images (500×) for the films fabricated with mv3glc (low WVP values). ^a–d^ data with the same letters are statistically similar at a 95% confidence level.

**Figure 8 ijms-25-06154-f008:**
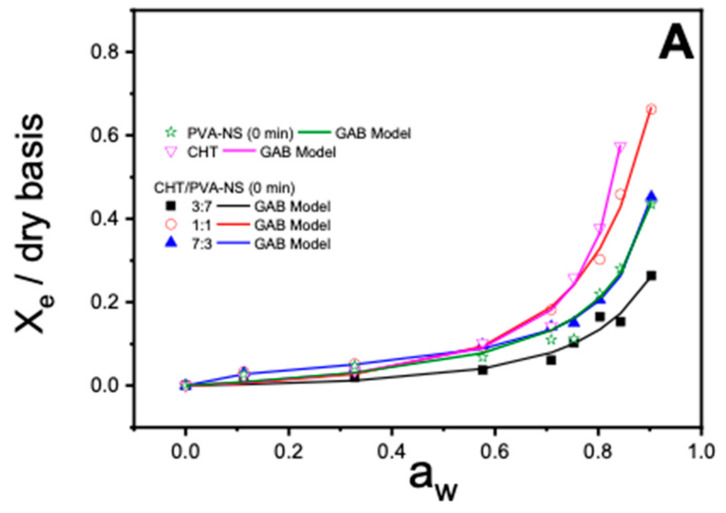
Equilibrium moisture sorption isotherms and respective fittings using the GAB model using: (**A**) non-sonicated PVA, PVA-NS; (**B**) 3 min of sonication, PVA_3_-NPs; (**C**) 5 min of sonication, PVA_5_-NPs; (**D**) 7 min of sonication, PVA_7_-NPs. Molar ratios for the CHT/PVA-NS or CHT/PVA_n_-NPs mixtures were 3/7, 1/1, and 7/3.

**Figure 9 ijms-25-06154-f009:**
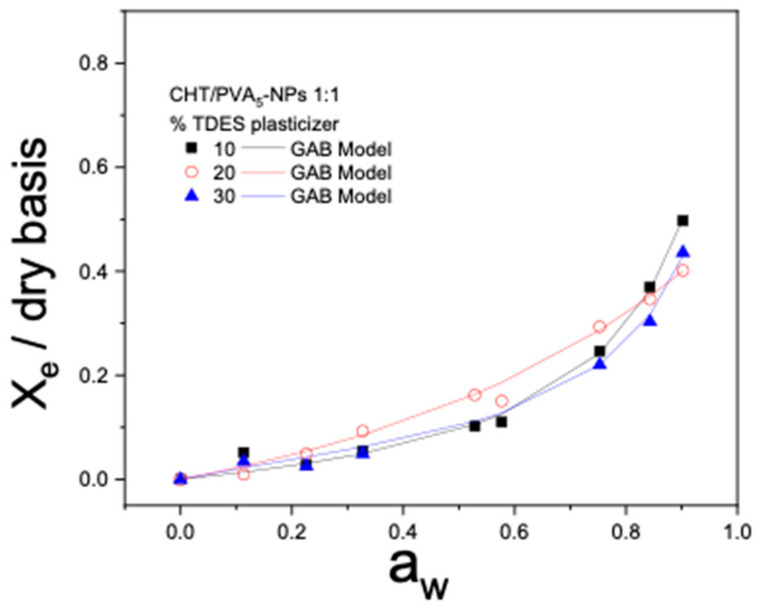
Equilibrium moisture sorption isotherm of the CHT/PVA_5_-NP 1:1 films formulated with 10 to 30% of the TDES plasticizer (*w*/*w*). The symbols are experimental data, and the lines are from the equations obtained by fitting the experimental data to the GAB equation.

**Figure 10 ijms-25-06154-f010:**
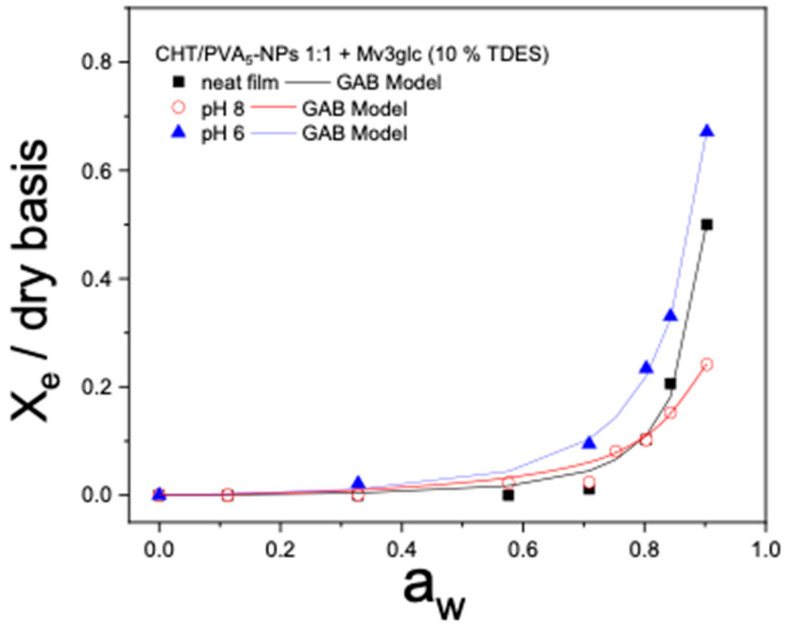
Experimental sorption data and respective fit using the GAB model for the CHT/PVA_5_-NP 1:1 film formulated with mv3glc (10% TDES) submitted at different pH conditions.

**Figure 11 ijms-25-06154-f011:**
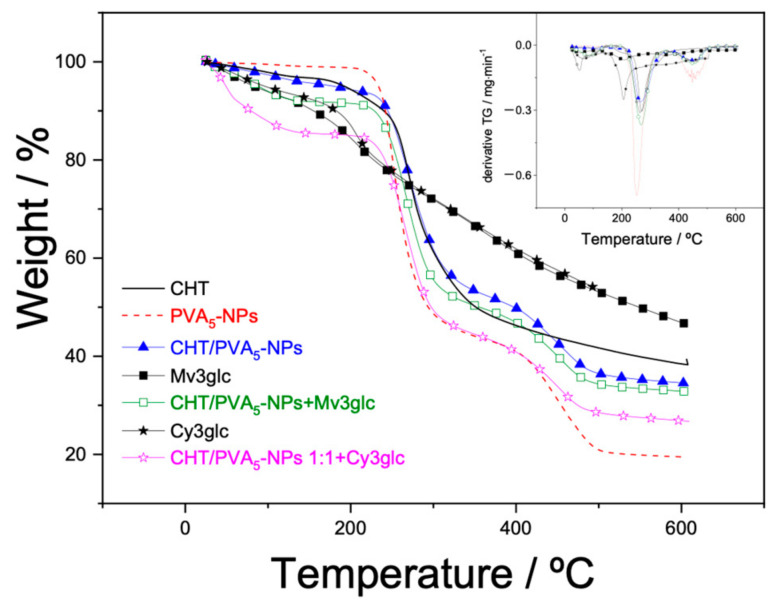
TG curves and DTG (inset graphic) curves for the thermal decomposition of anthocyanins (powder), and the films of CHT, PVA_5_-NPs, CHT/PVA_5_-NPs, and CHT/PVA_5_-NPs containing cy3glc and mv3glc.

**Table 1 ijms-25-06154-t001:** Refractive index values obtained for PVA, CHT, and DES samples.

PVA	Refractive Index
Sonication time (min)	nD
0	1.33565
1	1.33564
2	1.33565
3	1.33564
5	1.33564
7	1.33564
10	1.33564
13	1.33564
15	1.33565
CHT	1.33717
BDES (ChCl/Gly)	1.46578
TDES (ChCl/Gly/EG)	1.47776

**Table 2 ijms-25-06154-t002:** Intrinsic viscosity ([*η*]), and viscosity-average molecular mass (*M_v_*) of CHT in acetate buffer (0.25 M, pH 4.7).

Sample	[*η*] (dL·g^−1^) ^a^	[*η*] (dL·g^−1^) ^b^	*M_v_* (KDa)
CHT	4.35	4.29	258

^a^ from Huggins extrapolation. ^b^ from Kramer extrapolation.

**Table 3 ijms-25-06154-t003:** GAB parameters obtained from the fitting of the data displayed in Figure 8A–D to the GAB equation for the biobased films prepared from chitosan (CHT), non-sonicated PVA (PVA-NS), sonicated PVA (PVA_n_-NPs), and non-sonicated and sonicated blends (CHT/PVA-NS and CHT/PVA_n_-NPs).

	Ratio	*d* (mm) × 10^−2^	Solubility	C	*k*	*X* _0_
CHT	-	2.93 ± 0.21	26.92 ± 0.34	6.66	0.960	0.059
CHT/PVA-NS (0 min)	3:7	3.28 ± 0.14	20.76 ± 0.45	3.05	0.871	0.058
	1:1	3.27 ± 0.33	19.16 ± 0.70	4.10	0.955	0.095
	7:3	3.23 ± 0.63	21.58 ± 2.32	6.28	0.931	0.075
PVA-NS (0 min)	-	1.83 ± 0.26	20.74 ± 1.38	1.40	0.99	0.05
CHT/PVA_3_-NPs (3 min)	3:7	3.43 ± 0.05	21.04 ± 0.81	2.923	0.859	0.184
	1:1	2.56 ± 0.14	16.84 ± 3.30	4.34	0.970	0.097
	7:3	3.35 ± 0.35	24.15 ± 2.58	1.38	0.959	0.118
PVA_3_-NPs (3 min)	-	2.10 ± 0.03	19.99 ± 1.68	1.333	0.877	0.246
CHT/PVA_5_-NPs (5 min)	3:7	3.45 ± 0.42	14.87 ± 0.14	2.10	0.937	0.099
	1:1	3.82 ± 0.05	12.31 ± 0.18	1.08	0.890	0.164
	7:3	3.05 ± 0.65	10.85 ± 0.04	3.11	0.983	0.076
PVA_5_-NPs (5 min)	-	2.33 ± 0.14	17.47 ± 0.28	1.60	0.984	0.081
CHT/PVA_7_-NPs (7 min)	3:7	2.70 ± 0.65	26.80 ± 6.41	1.45	0.883	0.145
	1:1	3.09 ± 0.22	23.64 ± 0.59	1.29	0.947	0.115
	7:3	2.40 ± 0.04	25.10 ± 1.01	3.20	0.69	0.09
PVA_7_-NPs (7 min)	-	3.19 ± 0.06		1.82	1.00	0.059

**Table 4 ijms-25-06154-t004:** GAB parameters obtained from the fitting of the data displayed in Figure 9 to the GAB equation for the biobased films prepared from CHT/PVA_5_-NP (1:1) in the presence and absence of different % of TDES plasticizer.

	% TDES	*d* (mm) × 10^−2^	C	*k*	*X* _0_
CHT/PVA_5_-NPs 1:1	10	3.64 ± 0.46	1.12	0.908	0.109
	20	4.09 ± 0.30	0.145	0.386	3.641
	30	3.49 ± 0.50	2.99	0.922	0.077

**Table 6 ijms-25-06154-t006:** CIELAB color coordinates of the films (day 0) after being immersed in water at different pH values.

pH	Cross-Section (5000×)	Surface (10,000×)	Aspect	CIELAB Parameters(First Day of Preparation)
L*	a*	b*
3			solubilized	-	-	-
4	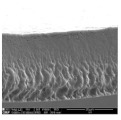	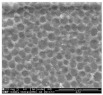	solubilized	-	-	-
5	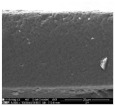	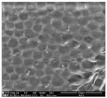		43.97 ± 1.2	26.25 ± 1.3	−6.42 ± 1.4
Water (5.7)	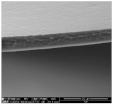	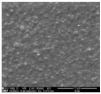	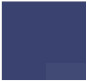	30.23 ± 1.1	9.52 ± 0.3	−25.26 ± 0.4
6	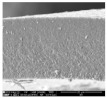	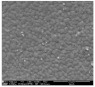		43.43 ± 2.1	20.65 ± 0.3	−12.12 ± 0.8
7	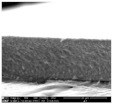	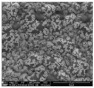	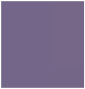	45.49 ± 0.8	10.66 ± 0.1	−16.92 ± 0.2
7.6	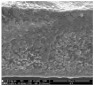	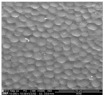	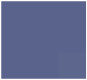	43.27 ± 1.5	7.00 ± 0.3	−21.81 ± 0.2
8	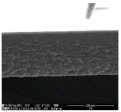	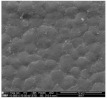		43.50 ± 1.1	2.82 ± 0.3	−25.36 ± 0.4
8.2	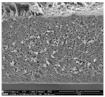	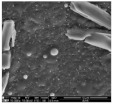	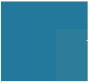	48.90 ± 0.5	−6.70 ± 0.1	−21.89 ± 0.2

**Table 7 ijms-25-06154-t007:** CIELAB color coordinates of the films measured over time conditioned in desiccators at different water activities (a_w_).

Film in Water (pH 5.7)
a_w_	Exposed Time/Days	Visual Easy RGB	CIELAB Coordinates	Exposed Time/Days	Visual Easy RGB	CIELAB Coordinates
L*	a*	b*	L*	a*	b*
	0	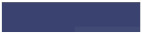	30.23 ± 1.1	9.52 ± 0.3	−25.26 ± 0.4					
0.1135	19		28.54 ± 2.1	5.48 ± 1.0	−23.05 ± 2.3					
0.0260		32.87 ± 1.8	2.38 ± 0.2	−23.84 ± 1.0					
0.3273		25.35 ± 1.0	8.47 ± 0.2	−23.76 ± 0.7					
0.5286		46.51 ± 2.5	5.32 ± 0.3	−22.06 ± 0.5					
0.5777		44.54 ± 3.4	6.11 ± 0.6	−20.50 ± 0.6					
0.7532		73.29 ± 0.4	1.47 ± 0.3	13.06 ± 0.6					
0.8432		75.15 ± 0.3	1.57 ± 0.2	13.53 ± 0.6					
0.9026		76.02 ± 0.3	1.59 ± 0.2	12.93 ± 0.5					
Film in Water (pH 6)
a_w_	Exposed Time/Days	Visual Easy RGB	CIELAB Coordinates	Exposed Time/Days	Visual Easy RGB	CIELAB Coordinates
L*	a*	b*	L*	a*	b*
	0		43.43 ± 2.1	20.65 ± 0.3	−12.12 ± 0.8					
0.1135	7	brittle (dry)	-	-	-	70	-	-	-	-
0.0260	brittle (dry)	-	-	-	-	-	-	-
0.3273	brittle (dry)	-	-	-	-	-	-	-
0.5286		50.10 ± 0.6	10.26 ± 0.2	−15.37 ± 0.2		56.11 ± 1.3	4.06 ± 0.2	−15.44 ± 0.2
0.5777		50.65 ± 1.8	8.44 ± 0.5	−10.32 ± 0.3		54.65 ± 1.9	4.37 ± 1.2	−13.61 ± 0.3
0.7532		35.52 ± 0.3	8.22 ± 0.3	−8.62 ± 0.6	shrunk	-	-	-
0.8432		51.55 ± 4.4	7.15 ± 0.5	−4.89 ± 0.6	shrunk	-	-	-
0.9026		56.41 ± 4.9	8.11 ± 0.4	0.88 ± 0.2	shrunk	-	-	-
Film in Water (pH 8.2)
a_w_	Exposed Time/Days	Visual Easy RGB	CIELAB coordinates	Exposed Time/Days	Visual Easy RGB	CIELAB Coordinates
L*	a*	b*	L*	a*	b*
	0	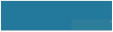	48.52 ± 0.5	−6.70 ± 0.1	−21.89 ± 0.2					
0.1135	7		51.49 ± 0.7	−6.48 ± 0.1	−20.83 ± 0.6	70		51.24 ± 0.8	0.65 ± 0.6	−23.50 ± 0.2
0.0260	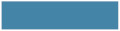	53.49 ± 0.9	−4.94 ± 0.3	−21.33 ± 1.2		55.65 ± 1.3	−3.94 ± 0.5	−20.94 ± 0.2
0.3273		36.70 ± 1.6	0.81 ± 0.1	−25.23 ± 1.5		34.82 ± 0.2	0.65 ± 0.1	−23.56 ± 0.4
0.5286		33.62 ± 0.3	1.74 ± 0.3	−17.18 ± 2.3		34.37 ± 0.3	0.60 ± 0.5	−19.24 ± 0.6
0.5777		51.09 ± 1.0	−2.41 ± 0.3	−17.05 ± 1.1		53.29 ± 1.7	−5.58 ± 0.5	−12.75 ± 0.2
0.7532		37.61 ± 1.3	0.04 ± 0.02	−17.65 ± 1.0		49.69 ± 1.6	−2.26 ± 0.4	−6.48 ± 0.2
0.8432		58.05 ± 1.9	−3.46 ± 0.5	−6.13 ± 1.7		77.49 ± 0.5	0.11 ± 0.2	16.50 ± 1.5
0.9026		51.15 ± 1.3	−4.87 ± 0.4	−14.67 ± 0.2	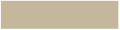	74.27 ± 0.9	−0.89 ± 0.2	15.22 ± 0.2

## Data Availability

The original contributions presented in the study are included in the article, further inquiries can be directed to the corresponding author.

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
