# Peer review of "Chitosan/Polyvinyl Alcohol-Based Biofilms Using Ternary Deep Eutectic Solvents towards Innovative Color-Stabilizing Systems for Anthocyanins"

_ijms, 2024, doi:10.3390/ijms25116154_

Round 1

Reviewer 1 Report

Comments and Suggestions for Authors

The study of de Souza et al. is interesting, but the paper needs revision before final decision regarding publication:

Lines 31-32: The statement needs revision.

The aim of the study should be clearly mentioned at the end of the Introduction section.

Lines 73: The procedure used for obtaining the malvidin-3-glucoside and cyanidin-3-glucoside should be presented in short.

Line 100: Renumbering of the 2.4.1. subsection should be reconsidered, as there is no 2.4.2 subsection.

Line 107: “Petri” should be capitalized.

Line 114: Rephrasing is needed. The main idea is difficult to understand.

Line 177: “Statistical Analysis” should be section 2.10.

Lines 197-198: Consider replacing “ … a “more” well-defined polymeric network.” by “ … a better defined polymeric network, compared to …”

Please check section 3.3. for the appropriate and consequent terminology. Is correct to used “refraction index” of refractive index”?

The abbreviations must be defined only at the first use in the manuscript body, and should be consequently used afterwards. See for instance “swelling index” at lines 69, 156, 157 and 254. The entire manuscript should be carefully check for other similar cases.

Figures 7a, 7b and 7c should be placed together. The results of the statistical analysis in this figure are not correct or complete. Please revise.

Line 340: Pay attention to the citation, which is not in agreement with the requirements of the journal.

Line 141: In a scientific paper, it is not allowed to state “Some authors have noted that  …” without clearly indicating the appropriate studies. Please revise.

Although the results are presented in a rather clear manner, in many cases the discussion against literature is missing.

The SD values and the statistical analysis results are not always provided. The authors should carefully revise the Results and discussion sections.

Some statements in the Materials and methods section should be revised, as the iThenticate software indicated a large text overlap (match) with the available literature.

Comments on the Quality of English Language

Minor English correction are needed.

Author Response

Reviewer: 1

The study of de Souza et al. is interesting, but the paper needs revision before final decision regarding publication:

Lines 31-32: The statement needs revision.

A: The sentence was reformulated.

The aim of the study should be clearly mentioned at the end of the Introduction section.

A: The main goal of the work was reinforced in the last paragraph of the manuscript.

Lines 73: The procedure used for obtaining the malvidin-3-glucoside and cyanidin-3-glucoside should be presented in short.

A: This information was added to the material and methods section.

Line 100: Renumbering of the 2.4.1. subsection should be reconsidered, as there is no 2.4.2 subsection.

A: The subsection 2.4.1. was deleted.

Line 107: “Petri” should be capitalized.

A: This was done.

Line 114: Rephrasing is needed. The main idea is difficult to understand.

A: The sentence was rephrased to avoid misunderstanding.

Line 177: “Statistical Analysis” should be section 2.10.

A: This was done.

Lines 197-198: Consider replacing “ … a “more” well-defined polymeric network.” by “ … a better defined polymeric network, compared to …”

A: The suggestion of the reviewer was followed and changed in the manuscript.

Please check section 3.3. for the appropriate and consequent terminology. Is correct to used “refraction index” of refractive index”?

A: To avoid ambiguities, the refractive index terminology was used.

The abbreviations must be defined only at the first use in the manuscript body, and should be consequently used afterwards. See for instance “swelling index” at lines 69, 156, 157 and 254. The entire manuscript should be carefully check for other similar cases.

A: We thank the reviewer’s comment and this was revised and changed accordingly throughout the entire manuscript.

Figures 7a, 7b and 7c should be placed together. The results of the statistical analysis in this figure are not correct or complete. Please revise.

A: Figures 7a, 7b and 7c were placed together. We thank the reviewer comment. The statistical analyses are correct. However, we forgot to include information regarding the statistical analyses. Therefore, to clarify the interpretation, the following information was added to the figure legend: “a-dSamples with the same letters in the same column are statistically similar at a 95% confidence level.”

Line 340: Pay attention to the citation, which is not in agreement with the requirements of the journal.

A: This was taken in account and changed accordingly.

Line 141: In a scientific paper, it is not allowed to state “Some authors have noted that  …” without clearly indicating the appropriate studies. Please revise.

A: A reference indicating those studies was added to this sentence.

Although the results are presented in a rather clear manner, in many cases the discussion against literature is missing.

A: The authors made an effort to add more references in the discussion section.

The SD values and the statistical analysis results are not always provided. The authors should carefully revise the Results and discussion sections.

A: The reviewer is absolutely correct. Not all results were analyzed statistically. The authors conducted the analysis only in situations they believed to be relevant. In the cases where the statistical analysis were conducted, information about the statistical analyses has been added to the text (or legend).

Some statements in the Materials and methods section should be revised, as the iThenticate software indicated a large text overlap (match) with the available literature.

A: This was done.

Reviewer 2 Report

Comments and Suggestions for Authors

The article “Chitosan-polyvinyl alcohol-based biofilms using ternary deep eutectic solvents towards innovative color-stabilizing systems for anthocyanins” presents results in general already studied in depth in previous works. The main novelty proposed is the use of PVA nanoparticles in the process. I think that the text should be improved before being published and describe in greater depth and reinforce the novelties of the work.

Extensive work has been carried out with the samples, presenting the results that the authors have considered to be of greatest interest, which I consider appropriate

Figures and legends. In general, it would be very convenient to make a more detailed and clarifying description of the figures, which favours the understanding of the results presented.

Based on what criteria the water immersion times and sonification times were decided for each type of experiments, I think they should be discussed in more detail in the text.

Comments:

Abstract and introduction

The novelty provided in this work must be reinforced in these sections.

2.3. Preparation of PVA nanoparticles

Please, incorporate references of the used method.

Line 93 Measurements were performed in triplicate. Measurements or maybe samples?

2.4. Preparation of solutions and 2.4.1, 2.5 and 2.6.

References to define the preparation method.

3.1

Figure 1. Define legend a, b and, A and B in the figure.

Figure 2. Define Lovis Dyn. Viscosity data presented.

Explain differences with Viscosity in mPa.s units in Figure 2 and Table 1 of reference 35.

Density variation is very low in Figure, it is difficult to stablish trends. But in Line 209-10, it is described in text 2 minutes show a density increase, significative from the point of view of the authors. Why also at 7 min?

3.3

Line 223-227 references

3.3 and 3.4. Any explanation about results?

3.6. Swelling Index (SI)

Figure 4. Describe legend for a best understood of the figure (for example 3/7, 1/1, 7/3).

Figure 5. Describe legend for a best understood of the figure.

Where are the results for 20 in absence of mv3glc in the figure?

Figure 6. Describe legend for a best understood of the figure.

Do the authors think it would be possible to prepare a table with the quantitative results for comparison?

3.7. Water Vapor Permeability (WVP)

Figure 7a and b. Describe legend for a best understood of the figures.

3.8. Moisture Sorption Isotherms

Figure 8, 9 and 10. Describe legend for a best understood of the figures.

3.9. Thermogravimetric analysis (TGA)

Figure 11 does not allow to see the results in its upper right part. The figure must be improved.

Conclusions

If the main novelty of the work is to produce nanoparticles to formulate with CHT dispersions, what are the advantages or disadvantages of this proposal?

What are the possible future uses of the new materials proposed?

Author Response

Reviewer: 2

The article “Chitosan-polyvinyl alcohol-based biofilms using ternary deep eutectic solvents towards innovative color-stabilizing systems for anthocyanins” presents results in general already studied in depth in previous works. The main novelty proposed is the use of PVA nanoparticles in the process. I think that the text should be improved before being published and describe in greater depth and reinforce the novelties of the work.

Extensive work has been carried out with the samples, presenting the results that the authors have considered to be of greatest interest, which I consider appropriate

Figures and legends. In general, it would be very convenient to make a more detailed and clarifying description of the figures, which favours the understanding of the results presented.

A: The caption of figures and tables were revised in order to better understand the results presented.

Based on what criteria the water immersion times and sonification times were decided for each type of experiments, I think they should be discussed in more detail in the text.

A: The sonication assays for the formation of PVA nanoparticles were decided to be studied over time and the data analyzed for each point. The water immersion experiments of the films was also studied over time and presented the most relevant data in the manuscript. These information’s was reinforced in the text.

Comments:

Abstract and introduction

The novelty provided in this work must be reinforced in these sections.

A: The main objectives and novelty were reinforced in these sections.

2.3. Preparation of PVA nanoparticles

Please, incorporate references of the used method.

A: The reference was added to this section.

Line 93 Measurements were performed in triplicate. Measurements or maybe samples?

A: The information was corrected. Samples were performed in triplicate.

2.4. Preparation of solutions and 2.4.1, 2.5 and 2.6.

References to define the preparation method.

A: There is no reference for section 2.4.1 because it was experimentally tested until get the best result. Section 2.5 is not referenced because this is the ordinary procedure to prepare films by solvent casting. Section 2.6 procedure is related with DLS equipment parameters which usually are not referenced.

3.1

Figure 1. Define legend a, b and, A and B in the figure.

A: The legends of letters a, b, c, A and B were defined in the figure caption.

Figure 2. Define Lovis Dyn. Viscosity data presented.

A: Lovis Dyn. Viscosity means Lovis Dynamic Viscosity.

Explain differences with Viscosity in mPa.s units in Figure 2 and Table 1 of reference 35.

A: The solutions were prepared under different conditions. On one hand, and according to reference 35 (in old version of the manuscript), the values obtained by the authors are higher than those obtained by us (PVA solutions prepared in acetate buffer). The higher viscosity of PVA solutions prepared in water compared to those in acetate buffer could be attributed to the stronger hydrogen bonding interactions and molecular entanglements facilitated by water as a solvent.

Density variation is very low in Figure, it is difficult to stablish trends. But in Line 209-10, it is described in text 2 minutes show a density increase, significative from the point of view of the authors. Why also at 7 min?

A: Thank you to the reviewer for the pertinent comment. The density values (along with the standard deviations, which are very small and not clearly visible in the image) show the trend described. We used 7 minutes (and other longer sonication times) to verify (confirm) the trend.

3.3

Line 223-227 references

A: The references about the refractive index were added.

3.3 and 3.4. Any explanation about results?

A: The studies conducted aimed to characterize the precursor solutions of the mixture

3.6. Swelling Index (SI)

Figure 4. Describe legend for a best understood of the figure (for example 3/7, 1/1, 7/3).

A: A legend describing the molar ratios was added to caption of figure 4.

Figure 5. Describe legend for a best understood of the figure.

A: A legend was reformulated to better understand the figure 5.

Where are the results for 20 in absence of mv3glc in the figure?

A: They are overlap by the results of 10% in absence of mv3glc.

Figure 6. Describe legend for a best understood of the figure.

A: A legend was reformulated to better understand the figure 6.

Do the authors think it would be possible to prepare a table with the quantitative results for comparison?

A: We thank the reviewer’s suggestion, however, we consider that a preparation of a table showing all the SI values could make the paper too heavy and difficult to follow the interpretation as well. Actually, Figures 4A-D are related with the SI study of all films formulated at different molar ratios and obtained after different PVA sonication periods; Figure 5 shows the effect of the presence of mv3glc on the SI of the film final formulation considering different content of TDES and Figure 6 compares the SI differences obtained with films containing a synthetic or a natural mv3glc-based dye.

3.7. Water Vapor Permeability (WVP)

Figure 7a and b. Describe legend for a best understood of the figures.

A: The letters a,b,c,d were defined in the figure caption: a-ddata with the same letters in the same column are statistically similar at a 95% confidence level.

3.8. Moisture Sorption Isotherms

Figure 8, 9 and 10. Describe legend for a best understood of the figures.

A: The captions of figures 8, 9 and 10 were reformulated to better understand the figures.

3.9. Thermogravimetric analysis (TGA)

Figure 11 does not allow to see the results in its upper right part. The figure must be improved.

A: The size of figure was increased to better see the results of the derivative TG Inset.

Conclusions

If the main novelty of the work is to produce nanoparticles to formulate with CHT dispersions, what are the advantages or disadvantages of this proposal?

What are the possible future uses of the new materials proposed?

A: The main advantages is that the PVA nanoparticles with 5 minutes of sonication revealed to be crucial for the fabrication of films with chitosan that allowed a great color stabilization of mv3glc over time especially for neutral to basic pHs where normally purple-blue quinoidal species are not thermodynamically stable. The stabilization of anthocyanins in soft and solid materials is foreseen as innovative systems for food industry namely to substituted synthetic additives for natural dyes and to potentiate their use as delivery and bioactive systems in food packaging.

Reviewer 3 Report

Comments and Suggestions for Authors

The manuscript entitled "Chitosan-polyvinyl alcohol-based biofilms using ternary deep eutectic solvents towards innovative color-stabilizing systems for anthocyanins" describes the use of PVA and chitosan-based films plasticized with DES for anthocyanin preservation. The work is quite current and the methods are adequately described. However, the paper has some critical issues that need to be addressed before publication. In the introduction, the section on DESs is completely missing. DESs are mixtures of compounds that have a negative melting temperature deviation from the ideal eutectic. The main feature of DESs is their tunability. In fact, by changing the nature of the components and their composition, it is possible to change the chemical and physical properties and adapt them to a specific application. (10.1016/j.molliq.2023.121563) Very interesting for the manuscript is the application of DESs in the extraction of anthocyanins. Some examples should be mentioned in the introduction. (10.1039/D4GC00526K, 10.1016/j.scp.2023.101168)

Although the characterization of the film is quite complete, a study of SDC to evaluate the Tg of the film may be of interest. In addition, a rheological study would also be useful to determine if gel formation occurs in the presence of water. From an application point of view, this would be a very interesting aspect. Only after these modifications can the paper be reconsidered for publication.

Author Response

Reviewer: 3

The manuscript entitled "Chitosan-polyvinyl alcohol-based biofilms using ternary deep eutectic solvents towards innovative color-stabilizing systems for anthocyanins" describes the use of PVA and chitosan-based films plasticized with DES for anthocyanin preservation. The work is quite current and the methods are adequately described. However, the paper has some critical issues that need to be addressed before publication. In the introduction, the section on DESs is completely missing. DESs are mixtures of compounds that have a negative melting temperature deviation from the ideal eutectic. The main feature of DESs is their tunability. In fact, by changing the nature of the components and their composition, it is possible to change the chemical and physical properties and adapt them to a specific application. (10.1016/j.molliq.2023.121563) Very interesting for the manuscript is the application of DESs in the extraction of anthocyanins. Some examples should be mentioned in the introduction. (10.1039/D4GC00526K, 10.1016/j.scp.2023.101168)

A: We thank the reviewer’s comment. The missing information about DES in the introduction section was added as well as the suggested references.

Although the characterization of the film is quite complete, a study of SDC to evaluate the Tg of the film may be of interest. In addition, a rheological study would also be useful to determine if gel formation occurs in the presence of water. From an application point of view, this would be a very interesting aspect. Only after these modifications can the paper be reconsidered for publication.

A: We appreciate the reviewer for the comment. The DSC studies have been conducted; however, the results were inconclusive. Therefore, we have decided not to include the results in the body of the email, but have provided them attached. The assays regarding rheological studies could be interesting, however, were not conducted due to the equipment’s unavailability.

Round 2

Reviewer 1 Report

Comments and Suggestions for Authors

The manuscript was improved, but some changes are still needed.

In my opinion the statistical analysis results are not provided where appropriate.

The results presented in Figure 7 (a) are not clear. For instance: red column (which changes values) is always CHT/PVA-NS 3:7? How is this possible? Moreover, the statistical analysis results presented in Figure 7 are nor complete or correct. It has no sense to me. Moreover, the text inserted in the caption is not clear: what the authors meant by “the same column”? The authors should carefully check these results.

Minor editing of English language is recommended.

Comments on the Quality of English Language

Minor editing of English language is recommended.

Author Response

Reviewer 1:

The manuscript was improved, but some changes are still needed.

In my opinion the statistical analysis results are not provided where appropriate.

The results presented in Figure 7 (a) are not clear. For instance: red column (which changes values) is always CHT/PVA-NS 3:7? How is this possible? Moreover, the statistical analysis results presented in Figure 7 are nor complete or correct. It has no sense to me. Moreover, the text inserted in the caption is not clear: what the authors meant by “the same column”? The authors should carefully check these results.

Minor editing of English language is recommended.

A: The figure 7 (a) was reformulated to include the caption on X axis to discriminate that the different set of columns are related with the time of sonication, from 0 min (non-sonicated samples) to 7 min. Regarding the statistical analysis, all the columns that do not include letters refer to statistically different results. To clarify, the figure's caption was rephrased to: a-ddata with the same letters are statistically similar at a 95% confidence level.

Reviewer 2 Report

Comments and Suggestions for Authors

The text has improved with the changes made

Author Response

We thank the reviewer comments

Reviewer 3 Report

Comments and Suggestions for Authors

I would like to thanks the author for accepting my suggestions. The manuscript is now suitable for publication.

Author Response

We thank the reviewer comments.

Round 3

Reviewer 1 Report

Comments and Suggestions for Authors

The authors answered to all questions.

Comments on the Quality of English Language

Minor editing of English language required